# Exploring the Impact of "Double Cycle" and Industrial Upgrading on Sustainable High-Quality Economic Development: Application of Spatial and Mediation Models

**Fayuan Wang** [1,2]**, Rong Wang** [3,]*** and Zhili He** [2]

1   Department of Economics & Management, Bozhou University, Bozhou 236800, China; wfy@yangtze.edu.cn
2   School of Economics and Management, Yangtze University, Jingzhou 434023, China; hzl@yangtze.edu.cn
3   Business School, Nanjing Xiaozhuang University, Nanjing 211171, China
*   Correspondence: rongwang@njxzc.edu.cn

**Abstract:** In the context of the integration into the world economy, the domestic and international cycles of the economy constitute the basic pattern of economic operation. This pattern is closely related to the industrial structure, and obviously it can affect quality of the economic development. The Chinese government has put forward an international and domestic dual cycle strategy for reinforcing an environmentally sustainable high-quality economic development. Similarly, it seems critically important to explore what impact the "dual cycle strategy" and industrial upgrading will have on the environmentally sustainable high-quality economic development. To do so, this paper adopted a dataset ranging from the year 2004 to 2019 from a regional perspective in China, and an empirical research is carried out based on the spatial Dubin model-mediation effect model to analyze the impact of "dual cycle strategy" and industrial upgrading on environmentally sustainable high-quality economic development. The results of the study show that the rising industrial structure effect has emerged prominently. Both the "dual cycle" and industrial structure upgrading have a positive spillover effect on environmentally sustainable high-quality economic development. Further, the direct effect of domestic circulation is significant, but the indirect effect is not obvious. In addition, we found that the direct and indirect effects of international circulation are both significantly positive; industrial upgrading has a partial mediating effect in the "double cycle". The study holds promising implications for a policy development.

**Keywords:** "dual cycle"; industrial upgrading; high-quality economy; Yangtze River Economic Belt

## 1. Introduction

Over the period of several decades, China has formed a pattern of a large economy with lower economic strength. To strengthen the economic situation, China formulated an environmentally sustainable high-quality economic development strategy, and proposed to emphasize over ecological priority, green development, and innovate the domestic and international "dual cycle" [1]. High-quality economic development requires getting rid of the traditional factor-dependent economic growth mode, relying more on the improvement of production efficiency and structural optimization brought about by technological innovation, and getting rid of excessive dependence on foreign demand [2]. It continues to promote circular economy, accelerate the transformation of economic development mode, build ecological civilization, promote green development, and achieve high-quality economic development [3,4].

In this context, China has proposed a dual cycle development pattern. The dual cycle is primarily based on the domestic large-scale cycle. Under the circumstance of high uncertainty in the external environment, the domestic industrial base is strong, the industrial chain is complete, and the strategic maneuvering space is large [5]. The characteristics of the super large market scale, smooth production, distribution, circulation, consumption,

and other aspects of economic operation promote the realization of internal self-circulation, including supply and demand circulation, industrial circulation, regional circulation, urban and rural circulation, and factor circulation [6]. Secondly, it is necessary to form a new pattern of international circulation, get rid of over-reliance on the traditional international circulation model [7], and form more international circulation patterns oriented towards southern countries and countries along the "Belt and Road". On the one hand, China, as a world market, has expanded its opening to the world and continued to share the opportunities of the Chinese market with the world; on the other hand, China, as the world's factory, will continue to provide the world with Made in China and Created in China, forming a new pattern of mutual promotion of domestic and international cycles [8]. This means that China's economic development has reached a stage where it must be transformed and upgraded.

The new dual cycle development pattern provides a very good idea for industrial upgrading, which is to continue to promote supply chain adjustment and industrial structure upgrade based on the domestic big market [9], and at the same time, based on the international perspective, international scope, and international market, to support Chinese enterprises to continue Focus on the world, expand opening up at a higher level, participate in the international division of labor at a deeper level, realize the optimized layout of the supply chain on a global scale, realize the transition of enterprises in the global value chain, and, finally, achieve high-quality economic development.

Therefore, we believe that this study can clearly explain its mechanism through the analysis of the spatial measurement section. Then, it is greatly important to clarify the role of industrial structure upgrading in the impact of international and domestic economic cycles on environmentally sustainable high-quality economic development that this study can fulfil. Further, although it is generally believed that international and domestic economic cycles can play a key to drive and promote environmentally sustainable high-quality economic development. However, economic cycles do not directly affect environmentally sustainable high-quality economic development and need to be coordinated with many factors such as the upgrading of industrial structure. In this regard, one can argue: to what extent does the upgrading of industrial structure play a role in it? Similarly, to solve such mysteries in the academic research, fresh evidence is critically important for the development of polices.

The purpose of this study is as follows: First, it is helpful to clarify the impact mechanism of international and domestic economic cycle and industrial structure upgrading on high-quality economic development, and to analyze the relationship between economic cycle, industrial structure and high-quality economic development in China. The second aspect is to test the effect of national and regional economic development. The role of implementing the national regional development strategy needs to be tested. This study is the first to test the effect of this strategy and provide experience for China to implement regional development strategies. The third point is to clarify the impact and mechanism of key elements in economic development on high-quality economic development. Through empirical testing, this study clarifies the mechanism of industrial structure upgrading in the international and domestic economic cycle and high-quality economic development, and provides inspiration for China to formulate economic development policies.

The arrangement of the study is as follows: the first part of paper is the introduction which explains research significance and contributions of this research. The Section 2 is the literature review, which is the core of the domestic and international research literature from different perspectives views and comment. Next, the third part is the research design, introducing the main research methods, the selection, and characteristics of measurement models. Following this, the fourth part is the empirical analysis, through the Moran index test, to determine the spatial model. Then, through the LM test, the spatial Dubin model is determined, and the direct and indirect effects of the core independent variables on the dependent variables are analyzed. Next, the mediation effect model is used to test the role of industrial structure upgrading in the dual cycle and high-quality economic

development. Finally, the last part draws conclusions and discusses several suggestions for a policy development.

## 2. Literature Review

Since the Chinese government put forward the "dual cycle" development strategy, "dual cycle" and high-quality economic development have become a hot topic in academic research and among the policy makers. In the past, various scholars have applied different methods and conducted research from different aspects and distinguished perspectives. In this section, we summarized the previous work in the following three aspects:

### 2.1. The Impact of "Dual Cycle" on High-Quality Economic Development

The "dual cycle" strategy is a scientific judgment made by the Chinese government with characteristics of international and domestic economic development. It is an active choice by the Chinese government to adapt to changes in the development stage, and it will inevitably become the normal state of economic development [10,11]. China's composition of a new "dual cycle" pattern is to completely break the institutional barriers that restrict the flow of factors to improve the regional linkage mechanism of production factors, and promote the cross-regional flow and optimal allocation of factors [12]. Similarly, it forms international and domestic economic cycles. In building domestic circulation system, the Chinese government has proposed consumer-side structural reforms, the essence of which is to promote consumption upgrading, to make it a new driving force for Chinese economic circulation [13,14].

Further, to construct a new development pattern of "dual cycles", strategically, we must persist in expanding domestic demand and consolidate the national economic cycle dominated by the domestic market [15]. The key to expanding domestic demand is to focus on promoting the upgrading of service consumption structure, forming an interactive integration of service utilization and product consumption, and driving the double cycles [15,16]. At the same time, by adjusting the economic structure to appropriately increase residents' consumption and effective investment through the formation of a sustainable growth mechanism for residents' income, a consumption-led domestic demand system needs to be established [16,17]. In addition, the key is also to adhere to the main domestic cycle and effectively respond to the impact of sudden changes in the international political and economic situation to maintain a good momentum of stable, healthy, and rapid growth of the national economy [18].

Based on this, this paper proposes:

**Hypothesis 1.** *"Dual circulation" can promote high-quality economic development in China.*

### 2.2. Industrial Upgrading Impact on High-Quality Economic Development

Following the COVID-19 global crises, one can argue that the world is facing major changes that were unseen a century ago. Similarly, the containment of China by developed countries has intensified, and international competition has now become more intense. In the new international situation, China must not only continue to maintain an important position in the global industrial chain, but also needs to further climb to the upstream of the global industrial chain, that is, to upgrade the industrial structure [19,20]. Industrial upgrading is manifested in economic development shifting from "quantity catch-up" to "quality catch-up", from "scale expansion" to "structural upgrading", from "factor-driven" to "innovation-driven", and from "distribution imbalance" to "common prosperity", from "high carbon growth" to "green development" [21–23]. Grossman and Krueger [24] researched that the level of resources and the environment is usually closely related to the stage of economic development. The environment of Yangtze River Economic Belt can be protected under the conditions of high-quality economic development [25]. Moreover, to certain extent it can be indicated that when a region implements high-quality economic development, its own good development conditions can attract capital, talents, technology,

and other factors to gather, and increase total factor productivity which is regarded as a high economic growth momentum of quality development [26,27]. Waqas et al. [28] found that big data analysis can improve the environmental performance of manufacturing, and Chinese manufacturing needs to formulate appropriate measures to address sustainability issues and promote the adoption of big data, which is of great significance for achieving high-quality development.

Based on this, this paper proposes:

**Hypothesis 2.** *Industrial upgrading can promote high-quality economic development in China.*

*2.3. "Dual Cycle" and Industrial Upgrading Impact on High-Quality Economic Development*

High-quality economic development ensures environmental sustainability which is driven by innovation. Similarly, to achieve environmentally sustainable economic growth, it is also pivotal that country should promote the effective allocation of economic resources among the industries and improve production efficiency according to consumer needs and promote the rational flow of labor among key industries through technological advancement [29]. In this regard, dual cycle strategy can provide basis for the development of economy and industry [30]. Similarly, constructing a new development pattern of "dual cycles" means to promote industrial upgrading through "dual cycles" to increase the efficiency of factor production and form new growth points [31]. Industrial upgradations adopt new industrial technology which fosters the initiative and first-mover advantage of the industry led by internal circulation, and creates an innovation ecosystem, which will drive high-quality economic development [32]. "Double cycle" and the synergy of industrial upgrading can realize the innovation chain as the leading factor in the "four chains" of innovation chain, supply chain, industrial chain, and value chain, and drive high-quality economic development [33]. Meanwhile, industrial upgrading could strengthen green technological innovation and green development, forming a mutually reinforcing pattern of ecological protection and high-quality development through ecological protection, and promoting the improvement of ecological protection through high-quality development [34,35].

Based on this, this paper proposes:

**Hypothesis 3.** *Industrial upgrading may play a mediating effect on high-quality economic development in the "dual cycle".*

From the existing research, it reflects the insights of scholars from different levels, and lays the foundation for further research on economic cycles and environmentally sustainable high-quality economic development. However, the Yangtze River Economic Belt, as a typical high-quality development demonstration area determined by the Chinese government, is how to improve and formulate various items in accordance with the internal functions and mechanisms of the "double-cycle" and high-quality development in the construction of a new international and domestic "double-cycle" development policy is a topic that requires fresh evidence through careful study. Starting from the reality of world economic development, this research combines the international and domestic "dual cycles" with industrial upgrading. It not only studies the effects of the "double cycles" and industrial upgrading on high-quality economic development, but also studies the international cycles, the impact of domestic circulation on high-quality economic development is the innovative point and unique value of this article compared to the existing research.

## 3. Material and Methods

### 3.1. Sample Description

China's Yangtze River Economic Belt includes regions/provinces of Shanghai, Jiangsu, Zhejiang, Anhui, Jiangxi, Hubei, Hunan, Chongqing, Sichuan, Guizhou, and Yunnan. Their

total economic output in 2019 accounted for 46.31%. Relying on the golden waterways of the Yangtze River and converging in the Yellow Sea and the East China Sea, these regions have natural advantages in economic development and foreign trade. It is a demonstration zone for the construction of a "dual cycle" of domestic and international economies. Therefore, taking these regions as the sample, and selecting the sample interval from 2004 to 2019, we will explore the impact of domestic and international "dual cycles" and industrial upgrading on high-quality economic development. The data collected from the National Bureau of Statistics and the provincial and municipal statistical bureaus, and the interpolation method is used to complement individual missing data.

### 3.2. Spatial Doberman Model

Compared with the ordinary panel model, the spatial measurement model can measure the spatial correlation and has numerous advantages. The spatial Dubin model (SDM) considers the spatial correlation between variables and is more widely used [36–38]. Therefore, following the recent papers, this paper selects SDM model to analyze the relationship among variables under investigation, the following spatial Dubin model is constructed:

$$HQED_{it} = \alpha_0 + \alpha_1 DEC_{it} + \alpha_2 LNIUP_{it} + \alpha_3 Col_{it} + \alpha_4 W_{ij} * DEC_{it} + \alpha_5 W_{ij} * LNIUP_{it} + \alpha_6 W_{ij} * Col_{it} + \mu_i + v_t + \varepsilon_{it}, \quad (1)$$

$$HQED_{it} = \beta_0 + \beta_1 DC_{it} + \beta_2 LNIUP_{it} + \beta_3 Col_{it} + \beta_4 W_{ij} * DC_{it} + \beta_5 W_{ij} * LNIUP_{it} + \beta_6 W_{ij} * Col_{it} + \mu_i + v_t + \varepsilon_{it}, \quad (2)$$

$$HQED_{it} = \gamma_0 + \gamma_1 IC_{it} + \gamma_2 LNIUP_{it} + \gamma_3 Col_{it} + \gamma_4 W_{ij} * IC_{it} + \gamma_5 W_{ij} * LNIUP_{it} + \gamma_6 W_{ij} * Col_{it} + \mu_i + v_t + \varepsilon_{it}. \quad (3)$$

In the above formula, $HQED_{it}$ means high-quality economic development, $DEC$ means domestic and international double circulation, $DC$ means domestic circulation, $IC$ means international circulation, LNST means industrial upgrading, LNFD is financial development (LNFD), financial investment (LNFI), industrial agglomeration (LNIG) the real economy (LNRD), and other control variables. $\alpha_i$, $\beta_i$, and $\gamma_i$ are the coefficients to be estimated, $\mu_i$ is spatial fixed effect, $v_t$ is time fixed effect, $\varepsilon_{it}$ is the error term, and $W_{ij}$ represents the spatial weight matrix (the same below). Regarding the spatial weight matrix, commonly used are 0–1 adjacency weight matrix (*W1*), geographic distance weight matrix (*W2*), and economic distance spatial weight matrix (*W3*). The specific forms are

$$W1_{ij} = \begin{cases} 1 & \text{When area } i \text{ is adjacent to } j \\ 0 & \text{other} \end{cases}, \quad (4)$$

$$W2_{ij} = \begin{cases} 1/l_{ij} & \&i \neq j \\ 0 & \&i = j \end{cases}, \quad (5)$$

$$W3_{ij} = \begin{cases} 1/|y_i - y_j| & \&i \neq j \\ 0 & \&i = j \end{cases}. \quad (6)$$

$l_{ij}$ is the Euclidean distance $i$ and $j$, $\overline{y_i}$ and $y_j$ are the GDP per capita differences between regions $i$ and $j$. In this study, *W2* is selected for correlation testing and the model is introduced, and *W1* and *W3* are used for robustness testing.

### 3.3. Mediating Effect Model

In the three-variable mediating effect, if variable X can have a certain effect on Y by influencing variable M, then M is mediating variable. Figure 1 shows a schematic diagram of the three-variable mediating effect.

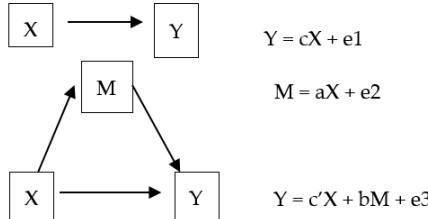

**Figure 1.** Schematic diagram of the intermediary effect of three variables.

As shown in Figure 1, ei represents the random disturbance term. When the regression coefficients a, b, and c in the model are significant at a certain test level, that will be exist mediating effect which accounts as ab/c. Otherwise, it is a complete mediation effect; if c′ is significant, it is an incomplete mediation effect, also called a partial mediation effect. This study is based on the stepwise regression method proposed by Baron and Kenny [39], Wen et al. [40], constructs model as follows:

$$HQED_{it} = \tau_0 + \tau_1 DEC_{it} + \tau_2 Col_{it} + \tau_3 W_{ij} * DEC_{it} + \tau_4 W_{ij} * Col_{it} + \mu_i + v_t + \varepsilon_{it}, \quad (7)$$

$$LNIUP_{it} = \varphi_0 + \varphi_1 DEC_{it} + \varphi_2 Col_{it} + \varphi_3 W_{ij} * DEC_{it} + \varphi_4 W_{ij} * Col_{it} + \mu_i + v_t + \varepsilon_{it}, . \quad (8)$$

In the above formula (Equations (7) and (8)), $\tau_i$ and $\varphi_i$ are the coefficients to be estimated, respectively. Obviously, combining Equations (1), (7) and (8), the coefficients to be estimated $\varphi_1$ and $\alpha_2$, $\tau_1$ and $\alpha_1$, respectively, which also represent a, b, c, and "c′" in the mediation effect model. Thus, Equations (1), (7) and (8) constitute a complete mediation effect test model.

*3.4. Variable Selection*

(1) Dependent variable: Environmentally sustainable high-quality economic development (HQED). The academic research has a wealth of study about connotation and evaluation indicators of high-quality economic development. Researchers have different understanding of connotation and the measurement indicators selected, the article believes that "innovation, coordination, green, open, and sharing" systematically summarizes the connotation of high-quality economic development [41]. Therefore, in this paper we select five dimensions to form environmentally sustainable high-quality economic development: coordination, greenness, openness, and sharing. The entropy method is used to objectively weight. Table 1 shows the comprehensive evaluation index system and the weight of each index.

In Table 1, environmentally sustainable high-quality economic development is embodied in the five dimensions, with weights of 0.2246, 0.2336, 0.1361, 0.1380, and 0.2676, respectively, and the distribution was not balanced. Specifically, the weights of innovation, coordination, and sharing are all greater than 0.22, while the weights of green and openness are both less than 0.14, and there is a big gap overall. This result shows that the current high-quality economic development of the Yangtze River Economic Belt focuses more on the sharing of public infrastructure, the coordination of economic growth efficiency and structural stability, and scientific and technological innovation. It is slightly weaker in terms of green environmental protection and opening to the outside world.

(2) Explanatory variables: domestic and international dual cycle (DEC) and industrial upgrading (IUP). The internal economic cycle is the process of achieving sustained growth in domestic production, distribution, exchange, and consumption through a certain path [42], while the economic external cycle drives economic development through exports. Based on this, a dual cycle comprehensive evaluation index system with 14 indicators in two dimensions is constructed. At the same time, the domestic circulation (DC) and the international circulation (IC) set as independent variables. The entropy method used for assigning weights to measure the domestic and international circulation of the Yangtze River Economic Belt. The results as follows Table 2.

**Table 1.** Comprehensive evaluation index system.

| Dimension Layer | Sub-Level | Index Layer | Unit | Attributes | Weights |
|---|---|---|---|---|---|
| Innovation (0.2246) | Innovation input | The proportion of science and technology expenditure in fiscal expenditure | % | + | 0.0546 |
| | | Full-time equivalent of R&D personnel | Person year | + | 0.0450 |
| | Innovation output | The ratio of technology market turnpover to GDP | % | + | 0.0568 |
| | | Number of domestic three types of patent applications granted per 10,000 people | item | + | 0.0682 |
| coordination (0.2336) | Industrial structure | Industrial Structure Upgrading Index | / | + | 0.0449 |
| | | Industrial Structure Theil Index | / | − | 0.0173 |
| | Financial structure | The ratio of deposit balance to GDP | % | + | 0.0404 |
| | | Ratio of loan balance to GDP | % | + | 0.0302 |
| | Growth fluctuations | Consumer Price Index | / | − | 0.0192 |
| | | Producer price index | / | − | 0.0227 |
| | | The absolute value of the fluctuation of the real GDP growth rate | / | − | 0.0179 |
| | Urban-rural structure | Per capita income ratio between urban and rural areas | / | − | 0.0186 |
| | | Per capita consumption ratio between urban and rural areas | / | − | 0.0223 |
| green (0.1361) | Environmental pollution | Wastewater discharge per unit GDP | Tons/ten thousand yuan | − | 0.0251 |
| | | Amount of industrial solid waste generated per unit of GDP | Tons/ten thousand yuan | − | 0.0147 |
| | Resources Consume | Energy consumption per unit of GDP | Tons of standard coal/ten thousand yuan | − | 0.0121 |
| | | Electricity consumption per unit GDP | KWh/yuan | − | 0.0136 |
| | Environmental protection | Green coverage rate in built-up area | % | + | 0.0185 |
| | | Industrial pollution control investment as a proportion of GDP | % | + | 0.0332 |
| | | Harmless treatment rate of domestic garbage | % | + | 0.0188 |
| open (0.1380) | Foreign trade dependence | The proportion of total import and export in GDP | % | + | 0.0764 |
| | Foreign investment | The actual utilization of foreign investment as a percentage of GDP | % | + | 0.0261 |
| | Foreign tourists | Number of international tourists received | Ten thousand people | + | 0.0355 |
| shared (0.2676) | Area sharing | Education expenditure as a share of GDP | % | + | 0.0412 |
| | | GDP per capita | Yuan/person | + | 0.0486 |
| | Urban and Rural Sharing | Urban registered unemployment rate | % | − | 0.0324 |
| | | Urbanization rate | % | + | 0.0360 |
| | Public Service | Medical and health expenditure as a proportion of GDP | % | + | 0.0335 |
| | | Highway mileage per capita | km/person | + | 0.0164 |
| | | Railway mileage per capita | km/person | + | 0.0174 |
| | | Number of medical and health institutions per 10,000 people | unit | + | 0.0203 |
| | | Public security expenditure as a proportion of fiscal expenditure | % | + | 0.0218 |

Note: the values in parentheses of each dimension are weights; the exchange rate used in the calculation of foreign trade and foreign investment indicators is the whole year of each year in China average RMB exchange rate; real GDP in growth fluctuations is calculated using 2002 as the constant base period, and its growth rate fluctuations are obtained by HP filtering.

**Table 2.** Domestic and international dual cycle comprehensive evaluation index system.

| Dimension Layer | Sub-Level | Basic Indicator Layer | Unit | Attributes | Weight a | Weight b |
|---|---|---|---|---|---|---|
| Domestic circulation (0.604) | produce (0.249) | Production price index | / | Reverse | 0.031 | 0.051 |
| | | Social productivity | Yuan/person | Positive | 0.101 | 0.167 |
| | | Real GDP per capita | yuan | Positive | 0.079 | 0.131 |
| | | Real GDP growth rate | % | Positive | 0.038 | 0.064 |
| | distribution (0.070) | Number of employees in the secondary and tertiary industries/total number of employees | % | Positive | 0.037 | 0.061 |
| | | Per capita income ratio of urban and rural residents | / | Reverse | 0.034 | 0.055 |
| | exchange (0.058) | Per capita retail sales of consumer goods | Yuan/person | Positive | 0.058 | 0.095 |
| | consumption (0.227) | Consumer Price Index | / | Reverse | 0.056 | 0.091 |
| | | Per capita consumption expenditure of urban residents | yuan | Positive | 0.103 | 0.166 |
| | | Per capita consumption expenditure of rural residents | yuan | Positive | 0.072 | 0.119 |
| International circulation (0.396) | trading (0.278) | Per capita consumption expenditure of rural residents | % | Positive | 0.121 | 0.310 |
| | | The ratio of total goods imports to GDP | % | Positive | 0.153 | 0.391 |
| | investment (0.118) | The ratio of foreign contracted project contract amount to GDP | % | Positive | 0.073 | 0.182 |
| | | The ratio of actual utilization of foreign investment to GDP | % | Positive | 0.045 | 0.117 |

Note: (1) Real GDP is calculated at constant prices in 2002; (2) the production price index and consumer price index are based on the chain index, and the fixed-base index with 2002 as the base period is calculated; (3) foreign exchange in trade and investment indicators use national average exchange rate for conversion; (4) the two indicators of distribution represent labor distribution and income distribution respectively; (5) the value in parentheses is the weight of each dimension in the dual cycle system; (6) the weight a is The weight of each indicator in the dual cycle integrated system, weight b is the weight of each indicator in the independent domestic cycle and the international cycle system.

The results in Table 2 show that the overall observation shows that the domestic and international "double cycles" are 0.604 and 0.396, respectively. It shows that the current economic development focuses more on the domestic economic cycle, and at the same time actively develops the international cycle. Specifically, in the "double cycle", the weights of the domestic cycle on production, distribution, exchange, and consumption are 0.249, 0.070, 0.058, and 0.227, respectively. The weight of production is the highest, followed by consumption, distribution, and the weight of exchange is lowest. When the domestic circulation is used as an independent variable, this same characteristic is also present. It shows that production and consumption are the most important parts in the current economic internal cycle of the region. The good interaction between production and consumption can speed up the domestic economic cycle and promote high-quality economic development. Regarding the international circulation, whether it is under the conditions of the "double circulation" comprehensive evaluation index system or the independent international circulation index measurement, the weight of trade is higher than that of investment. This shows that in the process of constructing a new domestic and international "dual cycle" pattern, the economic outer cycle of the region mainly depends on embedded import and export trade.

Regarding the specific indicators of industrial upgrading, refer to the existing literature [43] and choose to use the ratio of the added value of the tertiary industry to the added value of the secondary industry for measurement.

(3) Control variables: financial development (FD), financial support (FS), industrial agglomeration (IG), and real economy development (RD). With reference to related research, the proportion of financial industry added value, regional fiscal expenditure, industrial added value in GDP and GDP after excluding financial industry and real estate industry are selected respectively to measure [44–47].

Descriptive statistics were performed on the data before regression. We used the Stata15.0 measurement software to make descriptive statistics on each variable (Table 3), and the results were as follows.

**Table 3.** Descriptive statistics of raw data.

| Variable | Symbol | Number of Samples | Mean | Standard Deviation | Minimum | Maximum |
|---|---|---|---|---|---|---|
| High-quality economic development | HQED | 176 | 0.3791 | 0.1542 | 0.1946 | 0.7611 |
| Domestic and international double loop | DEC | 176 | 0.3211 | 0.2439 | 0.0325 | 0.9696 |
| Domestic circulation | DC | 176 | 0.3592 | 0.2356 | 0.0532 | 0.9439 |
| International circulation | IC | 176 | 0.2636 | 0.2688 | 0 | 1 |
| Industrial upgrading | IUP | 176 | 1.0152 | 0.3370 | 0.5985 | 2.6946 |
| Financial development | FD | 176 | 1409.121 | 1541.04 | 60.3 | 8158.23 |
| financial support | FS | 176 | 3920.857 | 2681.363 | 395.72 | 12,573.62 |
| Industrial Cluster | IG | 176 | 0.3799 | 0.0612 | 0.2147 | 0.5104 |
| Real economy | RD | 176 | 18,639.52 | 15131 | 1549.2 | 86,882.46 |

Specifically, the average value of the HQED is 0.3791, the standard deviation is 0.1542, and the maximum is 0.7611, the minimum is 0.1946, indicating that the sample values have obvious differences and large fluctuations. The domestic and international "dual cycles", domestic cycles, foreign cycles, industrial upgrades, and control variables all show the same characteristics, indicating that the level of HQED is obviously regional, with large gaps between different provinces and cities, and overall development. The imbalance problem is more prominent. Finally, take the natural logarithm of industrial upgrading and each control variable, and then introduce an empirical model to eliminate the influence of "heteroskedasticity".

## 4. Results and Discussion

We used the Eviews 9.0 software to analyze the data including descriptive statistical analysis and regression analysis. The analysis results are as follows:

### 4.1. Space Model

(1) Related inspection.

In the spatial measurement model, the spatial correlation of each variable is the prerequisite for the construction of the spatial model, and the Moran index (I) is usually used to test the spatial autocorrelation. We employed said model and the results as follows Table 4.

**Table 4.** Global Moran Index I of the main variables.

| Year | HQED | DEV | DC | IC | IUP |
|---|---|---|---|---|---|
| 2004 | 0.148 *** | 0.156 *** | 0.173 *** | 0.118 *** | 0.117 *** |
| 2005 | 0.138 *** | 0.161 *** | 0.176 *** | 0.125 *** | 0.110 ** |
| 2006 | 0.173 *** | 0.185 *** | 0.205 *** | 0.140 *** | 0.105 ** |
| 2007 | 0.180 *** | 0.170 *** | 0.182 *** | 0.131 *** | 0.094 ** |
| 2008 | 0.177 *** | 0.158 *** | 0.184 *** | 0.101 *** | 0.082 ** |
| 2009 | 0.204 *** | 0.155 *** | 0.185 *** | 0.093 *** | 0.072 ** |
| 2010 | 0.146 *** | 0.152 *** | 0.191 *** | 0.065 *** | 0.075 ** |
| 2011 | 0.178 *** | 0.145 *** | 0.186 *** | 0.055 *** | 0.061 ** |
| 2012 | 0.216 *** | 0.159 *** | 0.187 *** | 0.083 *** | 0.044 * |
| 2013 | 0.215 *** | 0.162 *** | 0.182 *** | 0.085 *** | 0.033 * |
| 2014 | 0.217 *** | 0.157 *** | 0.184 *** | 0.060 *** | 0.025 * |
| 2015 | 0.241 *** | 0.168 *** | 0.181 *** | 0.089 *** | 0.015 |
| 2016 | 0.262 *** | 0.182 *** | 0.184 *** | 0.124 *** | 0.018 * |
| 2017 | 0.242 *** | 0.201 *** | 0.183 *** | 0.170 *** | 0.013 |
| 2018 | 0.214 *** | 0.191 *** | 0.200 *** | 0.143 *** | 0.033 * |
| 2019 | 0.241 *** | 0.182 *** | 0.210 *** | 0.089 *** | 0.004 |

Note: * $p < 0.1$, ** $p < 0.05$, *** $p < 0.01$.

In Table 4, from 2004 to 2019, the global Moran index I of HQED, domestic and international "dual cycles", domestic cycles, and international cycles are all significantly positive. Moran index has only been insignificant for three years. This result means the core variables of the study have a significant positive spatial correlation, and the interdependence between provinces and cities is significant. Therefore, to explore the impact of domestic and international "double cycles" and technological progress on HQED, it is very appropriate to choose to build a spatial measurement model.

Before further constructing a specific spatial measurement model, some related tests used to select optimal model. Test results as follows Table 5.

**Table 5.** Test results related to model selection.

| Inspection Type | Null Hypothesis | Significance | Result |
|---|---|---|---|
| *LM* test | *SEM* model | 2.933 * | *SDM* model |
| | *SEM* model (steady) | 6.192 ** | |
| | *SAR* model | 8.176 *** | |
| | *SAR* model (steady ) | 11.434 *** | |
| *Hausman* test | Random effect | 372.66 *** | Fixed effect |
| *Wald* test | *SEM* or *SAR* model is better than *SDM* model | 13.56 ** | *SDM* model |
| *LR* test | *SEM* model is better than *SDM* model | 11.98 * | *SDM* model |
| | *SAR* model is better than *SDM* model | 12.53 * | *SDM* model |

Note: * $p < 0.1$, ** $p < 0.05$, *** $p < 0.01$.

First, the LM test method is used to determine the specific choice of SDM model, SEM model, or SAR model. The statistical results rejected the null hypothesis at different test levels, so the SDM model was chosen. Second, the Hausman test show that the result passed the significance test, so we chose to construct a fixed-effects model. Then, through Wald test and LR test, it is judged whether the SDM model can be reduced to SAR model or SEM model. It can be seen that results have passed different test levels to significantly, which further shows that it is more appropriate to construct an SDM model. Finally, considering the characteristics of the data in this study, choose to build SDM model with fixed time-point effects.

(2) Spatial model analysis.

To explore the impact of domestic and international double cycles and industrial upgrading on HQED, stata15.0 software was used to perform maximum likelihood estimate method on model (1). Results are presented in the Table 6.

**Table 6.** Estimation results of SDM model.

| Variable | Coefficient | z Statistics | Variable | Coefficient | z Statistics |
|---|---|---|---|---|---|
| *DEC* | 0.4535 *** | 10.71 | *W * LNIUP* | 0.3594 ** | 2.04 |
| *LNIUP* | 0.1391 *** | 3.29 | *W * LNFD* | −0.0459 | −0.79 |
| *LNFD* | 0.0342 *** | 2.84 | *W * LNFS* | 0.0646 | 0.61 |
| *LNFS* | 0.0500 * | 1.80 | *W * LNIG* | 0.2507 | 1.14 |
| *LNIG* | 0.2218 *** | 4.39 | *W * LNRD* | −0.0225 | −0.32 |
| *LNRD* | −0.0681 | −4.26 | $\sigma^2$ | 0.0009176 *** | 9.64 |
| *W * DEC* | 0.5692 *** | 2.92 | $R^2$ | 0.4920 | |
| | | | *log−likelihood* | 357.5565 | |

Note: * $p < 0.1$, ** $p < 0.05$, *** $p < 0.01$.

Table 6 shows that the model goodness of fit $R^2$ is 0.4920, indicating that the overall fitting ability is better, and the results are accurate. According to specific analysis, the coefficient of the domestic and international "dual cycle" is 0.4535, indicating that the "dual cycle" can significantly promote HQED, Hypothesis 1 has been verified, and scholars Wang and Niu [12] have also reached the same conclusion. They believe that the new development pattern of "dual circulation" not only injects new vitality into the international cycle, but also internal and external interaction, two-way linkage, dynamic In the process of change,

the domestic market and the international market are constantly interacting, promoting each other, and integrating into one, and promoting the development of higher quality, higher level and higher level of the economy. The coefficient of *W * DEC* is 0.5692, and it has passed the 1% significance level, indicating that the domestic and international "dual cycle strategy" of this province and city also has a significant role in promoting the HQED of neighboring provinces and cities. This result shows that the regional economic development in the Yangtze River Economic Belt, the domestic and international "dual cycles" play an important role in promoting HQED. The connections and interactions between different provinces and cities are close, and the overall benefits continue to be better. The benefits of "double circulation" have become increasingly prominent.

The coefficient of industrial upgrading is 0.1391, indicating that industrial upgrading can significantly promote HQED, Hypothesis 2 has been verified, which is consistent with the conclusions drawn by most scholars. For example, Peneder [48] pointed out that the optimization and upgrading of the industrial structure can achieve the improvement of the overall social productivity by regulating the continuous flow of production factors among industries with different production efficiencies, resulting in "Structural dividends" that drive economic growth. Wu et al. [49] calculated the degree of industrial structure change through Moore index and found that active industrial structure optimization and adjustment can significantly improve total factor productivity. The coefficient of W* LNIUP has passed the 5% significance level test, and the coefficient is 0.3594, indicating that the positive spatial spillover effect of industrial upgrading in these regions. Industrial upgrading has realized the structural transformation and technological upgrading of traditional industries, can promote the rational allocation of economic system resources, and promote HQED.

Since the implementation of China's Yangtze River Economic Belt development strategy, the Yangtze River as a whole region have continued to strengthen cooperation and exchanges, continuously deepen and practice the concept of co-construction and sharing, and continue to promote the rationalization and advancement of the entire industrial chain layout, which will drive the high-quality economic development of all region.

For further exploring the impact of the domestic and international "dual cycles" on the HQED in the Yangtze River Economic Zone of China, the domestic and international cycles are regarded as independent variables for observation, and model (2) and model (3) are introduced for estimation. Results as follows Table 7.

**Table 7.** Estimated results of domestic circulation and international circulation models.

| Variable | HQED Model (2) | HQED Model (3) | Variable | HQED Model (2) | HQED Model (3) |
|---|---|---|---|---|---|
| *DC* | 0.4091 *** | / | *W * DC* | 0.3877 | / |
|  | 8.78 | / |  | (1.61) | / |
| *IC* | / | 0.3489 *** | *W * IC* | / | 0.4948 *** |
|  | / | (10.03) |  | / | (3.40) |
| *LNIUP* | 0.1974 *** | 0.1708 *** | *W * LNIUP* | 0.6279 *** | 0.2987 * |
|  | (4.56) | (4.07) |  | (3.29) | (1.72) |
| *Col* | √ | √ | *log-likelihood* | 344.5641 | 351.7626 |
| $\sigma^2$ | 0.001048 *** | 0.000983 *** | $R^2$ | 0.3926 | 0.4284 |

Note: * $p < 0.1$, *** $p < 0.01$.

As shown in the results in Table 7, in model (2), the coefficient of domestic circulation is 0.4091, indicating that the internal economic circulation can significantly promote high-quality economic development. The coefficient of *W * DC* is positive and not significant, which means that the economic internal circulation of a certain province has no obvious positive effect on the high-quality economic development of neighboring provinces and cities.

In model (3), the coefficients of international circulation and *W * IC* are 0.3489 and 0.4948, respectively, which are both significantly positive, indicating that economic external

circulation can promote the HQED of the province and city, and it has a significant positive spillover effect.

The above results indicate that during the research period, the regional economic development adheres to the development strategy of parallel domestic and international cycles, from consumption, production, distribution, and exchange of the domestic economy to investment and trade in the international economy, all-round service for HQED. However, in terms of spatial spillover effects, there is still a gap between domestic circulation capabilities and international circulation; domestic circulation should become the focus of the future "dual circulation" construction.

Regarding industrial upgrading, whether it is in model (2) or model (3), whether it is a direct effect or an indirect effect, it is significantly positive to varying degrees. This result once again shows that China's Yangtze River Economic Belt development strategy has achieved remarkable results, and the overall benefits of regional industrial upgrading have been formed.

It can be seen from this that there are many studies on dual circulation, industrial upgrading and high-quality economic growth, but few analyze specific regions, and a large number of scholars are qualitative analysis, there are few quantitative analyses, and there is a lack of research on the quantitative relationship between variables; in addition, no scholars' research in this field has been found on the spatial spillover effects of validation variables. This paper is a useful addition to existing research on analyzing its spillover effects.

(3) Robustness test.

By introducing the 0–1 distance weight matrix (*W*1) and the economic distance weight matrix (*W*3), the robustness test of the model (1) is carried out to observe whether there are significant differences in the estimated results of each variable under the influence of different spatial matrices (the results are shown in Table 8).

**Table 8.** Model robustness test.

| Variable | *W*1 | | | *W*3 | | |
|---|---|---|---|---|---|---|
| | **Model (1)** | **Model (2)** | **Model (3)** | **Model (1)** | **Model (2)** | **Model (3)** |
| *DEC/DC/IC* | 0.4105 *** | 0.3248 *** | 0.3226 *** | 0.4654 *** | 0.5573 *** | 0.3053 *** |
| | (8.89) | (7.02) | (8.54) | (17.53) | (14.45) | (14.09) |
| *W * DEC/W * DC/W * IC* | 0.5082 *** | 0.2116 * | 0.5396 *** | 0.2416 ** | 0.1408 | 0.1671 |
| | (0.32) | (1.68) | (5.74) | (1.99) | (1.26) | (1.51) |
| *LNIUP* | 0.0756 * | 0.1625 *** | 0.0781 * | −0.0014 | −0.0101 | 0.0504 * |
| | (1.84) | (4.14) | (1.93) | (−0.05) | (−0.36) | (1.66) |
| *W * LNIUP* | −0.0543 | 0.0582 | −0.0344 | 0.0480 | 0.0457 | 0.0308 |
| | (−0.72) | (0.73) | (−0.47) | (0.83) | (0.72) | (0.46) |
| Control variable | √ | √ | √ | √ | √ | √ |
| *Log-Likelihood* | 368.4327 | 357.7823 | 365.5264 | 382.3697 | 373.5205 | 361.2863 |
| $R^2$ | 0.4582 | 0.3597 | 0.0116 | 0.3891 | 0.3543 | 0.3942 |
| $\sigma^2$ | 0.00089 *** | 0.00099 *** | 0.00091 *** | 0.00070 *** | 0.00084 *** | 0.00094 *** |

Note: Models (1), (2), and (3) are respectively the robustness test of the double cycle, domestic cycle and international cycle affecting HQED. * $p < 0.1$, ** $p < 0.05$, *** $p < 0.01$.

Robustness test results mean that the estimation results under the conditions of introducing *W*1 and *W*3 are only slightly different from the estimation results of *W*2 in the individual estimation coefficients. This result proves the rationality of the estimation method and the reliability of the empirical results.

*4.2. Mediation Effect Test*

(1) Intermediary effect test.

Estimation of model (7) and model (8), combined with model (1), empirically test the mediating effect of industrial upgrading in the domestic and international dual cycle impact on HQED (Table 9).

**Table 9.** Test results of mediation effect.

| Variable | *HQED* Model (1) | *HQED* Model (7) | *LNIUP* Model (8) |
|---|---|---|---|
| *DEC* | 0.4535 *** (10.71) | 0.5519 *** (17.94) | 0.6514 *** (10.88) |
| *LNIUP* | 0.1391 *** (3.29) | | |
| *W * DEC* | 0.5692 *** (2.92) | 0.9743 *** (6.14) | |
| *W * LNIUP* | 0.3594 ** (2.04) | | 1.8417 *** (6.75) |
| *Col* | √ | √ | √ |
| $R^2$ | 0.4920 | 0.1841 | 0.1765 |

Note: ** $p < 0.05$, *** $p < 0.01$.

As shown in Table 9, in the model (7), the coefficient of the domestic and international "dual cycle" is 0.5519 (c), which has passed the 1% test level, indicating that the "dual cycle" has significantly promoted HQED. In the model (8), the coefficient of the "double cycle" is 0.6514 (a), which is significant, indicating that the "double cycle" can significantly promote industrial upgrading. Moreover, in model (1), the coefficients of "double cycle" and industrial upgrading are 0.4535 ("c'") and 0.1391 (b), respectively, which both pass the 1% significance level. The result means that industrial upgrading plays a part of the mediating effect in the impact of the domestic and international "dual cycle" on HQED, accounting for 16.42%, Hypothesis 3 has been verified. In the research on the mediating effect of industrial upgrading, a large number of scholars focus on the relationship between industrial upgrading and economic growth, and fewer scholars in the dual cycle background. Therefore, the research in this paper provides a theoretical reference for existing research.

(2) Robustness test.

A basic regression model is used to build a robustness test of the mediating effect of industrial upgrading in the domestic and international "dual cycle" affecting the high-quality economic development (Table 10).

**Table 10.** Robustness test of mediation effect.

| Variable | Explained Variable | | |
|---|---|---|---|
| | HQED | HQED | LNIUP |
| DEC | 0.4668 *** (12.88) | 0.4988 *** (18.20) | 0.3424 *** (2.61) |
| LNIUP | 0.1041 *** (2.72) | | |
| Col | √ | √ | √ |
| $R^2$ | 0.6150 | 0.5943 | 0.9336 |

Note: *** $p < 0.01$.

As shown in the results in Table 10, in the intermediary effects of the three variables of domestic and international "dual cycles", industrial upgrading and HQED, the values of a, b, c, and "c'" are 0.3424, 0.1041, 0.4988 and respectively 0.4668, both are significant at the 1% test level, indicating that the analysis results of the entire spatial Dubin model are robust.

## 5. Conclusions

By constructing a domestic and international "dual cycle" and high-quality economic development index system and applying the spatial Dubin model to analyze the impact of "dual cycles" and industrial structure upgrading on economic high-quality development, the following basic conclusions can be drawn.

(1)  The regional construction effect of China's Yangtze River Economic Belt has been highlighted, and the domestic and international "dual cycle" has a significant positive effect on high-quality economic development. It indicates that overall, the "dual cycle" of China's Yangtze River Economic Belt can significantly promote high-quality economic development and has a significant indirect effect. It shows that not only the domestic and international "dual cycles" can significantly promote high-quality

economic development from the entire region, but also the domestic and international "dual cycles" of the provinces and cities can also significantly promote the high-quality economic development of neighboring provinces and cities. The reason is that the country has achieved remarkable results in implementing the development strategy of the regions, and close contacts and interactions have been formed between different provinces and cities in the entire region. The new "dual cycle" pattern has benefits for high-quality economic development increasingly prominent.

(2) The effect of upgrading the industrial structure of China's Yangtze River Economic Belt is apparent, which can significantly promote HQED and produce spillover effects. This shows that the regional industrial upgrading of Yangtze River Economic Zone not only promotes high-quality economic development, but also formed a positive spatial spillover effect. The reason for this is that in recent years, the State Council and local governments have taken strong measures to implement the "overall protection without major development" and achieved remarkable results, realizing the importance of industrial upgrading. Similarly, industrial upgrading has promoted the structural transformation of traditional industries. Furthermore, technological upgrades have effectively promoted the rational allocation of resources in the economic system, thereby comprehensively driving the optimization and upgrading of the industrial chain, and promoting HQED.

(3) The direct effect of the domestic circulation on HQED is significant, but the indirect effect is not yet obvious. It indicates that the economic internal circulation of Yangtze River Economic Belt region has significantly promoted high-quality economic development. Meanwhile, the coefficient of $W * DC$ is positive but not significant, indicating that the economic internal circulation of a certain province has no obvious positive impact on the high-quality economic development of neighboring provinces and cities. There are two main reasons for this: First, the provinces and cities earnestly implement the various policies of the Chinese government on the development of the Yangtze River Economic Belt, which has promoted the optimization and upgrading of the industrial structure of the provinces and cities, and promoted the high-quality economic development of the region; the implementation of the development plan is in progress, and the overall coordination of industrial layout and economic operation is still being formed. Therefore, the impact of domestic circulation on HQED is not significant.

(4) The direct and indirect spillover effects of the international circulation on HQED are both significantly positive. It shows that provinces and cities have significant positive spillover effects. For this reason, the regional economic development of China's Yangtze River Economic Belt has well implemented the development strategy of adhering to the parallel of domestic and international cycles, from consumption, production, distribution and exchange of the domestic economy to investment and trade in the international economy. Serving high-quality economic development has had a significant positive impact on high-quality economic development. In terms of spatial spillover effects, the level of international circulation is higher than that of domestic circulation, domestic circulation should become the focus of the future "dual circulation" construction.

(5) The industrial upgrading has played a part of the intermediary effect on HQED in the "dual cycle". The industrial upgrading constitutes an important link in the industrial chain, and has played a part of the intermediary role in the HQED in the domestic and international "dual cycle". This result further show that China's Yangtze River Economic Belt development strategies have achieved significant results, and regional industrial upgrading have achieved significant results, and the overall benefits for high-quality economic development have been formed. Meanwhile, HQED is a comprehensive manifestation of the level of economic development and is affected by the entire industrial chain. The domestic and international "dual cycle" has just formed a complete industrial chain, and industrial upgrading is an important link in

the industrial chain. The "dual cycle" has played a part of the mediating effect in the impact of the HQED.

In order to improve the high-quality economic development level of the Yangtze River Economic Belt, the author puts forward the following suggestions: First, the construction of a multi-level capital market and the development of a reasonable and orderly financial industry can help companies reduce financing costs, control and avoid business risks, and enhance market competitiveness, to promote industrial upgrading and transformation, and promote the cultivation of emerging industries; second, it is necessary to strengthen the training of domestic talents and establish a mechanism for the introduction of foreign high-end talents, so that talents must be used, and dedicated personnel are dedicated; third, make full use of new-generation information technologies such as big data Integrate industrial development with the digital economy, take industrial digitization as the core, and further promote the upgrading of the industrial structure; fourth, continue to promote circular economy, accelerate the transformation of economic development mode, build ecological civilization, promote green development, and achieve high-quality economic development; fifth, actively improve the rationalization level of the industrial structure and promote the optimization of the industrial structure. At the same time, on this basis, further enhance the advanced level of the industrial structure, promote the upgrading of the industrial structure, and realize a positive interaction between the two; sixth, strengthen the construction of transportation infrastructure and build a comprehensive Open channel. Build a three-dimensional transportation system such as railways, highways, aviation, and water transportation, and create a three-dimensional logistics and transportation network that connects the east and the west and connects the north and the south. Give full play to the advantages of water, land and air composite transportation, create an all-round development channel for the Yangtze River Economic Belt, and improve transportation accessibility; seventh, coordinate all aspects of high-quality economic development, and strive to build a scientific management system and mechanism for high-quality economic development based on actual conditions, and continuously improve the high-quality economic development level of cities in the Yangtze River Economic Belt; eighth, closely follow the national "dual circulation" strategy, strive to promote the transformation of growth momentum to rely more on domestic demand, promote the transformation of the industrial system to an independent and controllable, and promote the industrial system to be independent and controllable, and to promote the transformation of trade functions to agglomeration of global factors.

This study verifies the impact of "dual circulation" and industrial upgrading on high-quality economic development, but there are still deficiencies in the following aspects:

(1) Due to the limitation of research materials, the construction of the "dual circulation" index system may not be perfect, and various index systems can still be tried for analysis. The author will explore different indicators in further research to verify and compare with the research in this paper, expecting to draw more valuable conclusions.

(2) The high-quality economic development should also be affected by the national macro-policy. Since it is difficult to quantify the policy effect, this study has not yet covered it. In further research, we can explore how to quantify the qualitative indicators. In the case of adding the national macro-policy, analyze the spatial effect of high-quality economic development.

(3) For the measurement of high-quality and dual cycle indicators, due to the lack of existing literature, the main reference is to the documents issued by the state for condensed, with a certain degree of subjectivity. We hope to continue to summarize in future research and build objective and scientific indicators system.

**Author Contributions:** Conceptualization, Z.H.; methodology, F.W.; software, F.W.; validation, R.W., formal analysis, Z.H.; investigation, R.W.; resources, F.W.; data curation, F.W.; writing—original draft preparation, R.W.; writing—review and editing, R.W.; visualization, F.W.; supervision, Z.H.; project administration, F.W.; funding acquisition, F.W. All authors have read and agreed to the published version of the manuscript.

**Funding:** The paper was supported by The General Project of the National Social Science Foundation "Research on Promoting the High-quality Development of Manufacturing in the Yangtze River Economic Zone" (19BJL061).

**Institutional Review Board Statement:** Not applicable.

**Informed Consent Statement:** Not applicable.

**Data Availability Statement:** The data used to support the findings of this study are available from the corresponding author upon request.

**Conflicts of Interest:** The authors declare no conflict of interest.

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
