# Peer review of "Exploring the Impact of “Double Cycle” and Industrial Upgrading on Sustainable High-Quality Economic Development: Application of Spatial and Mediation Models"

_sustainability, doi:10.3390/su14042432_

Round 1
Reviewer 1 Report
Despite its frequent use of the term, the paper is not clear on what double-cycle development means. It appears to refer to domestic and international markets, but the paper should make this clear. The paper refers to industrial upgrading through the double cycle, but it is unclear what this means. Is it anything more than saying industry should be upgraded to meet demand? Some of that demand is domestic; some of it is foreign.
As an example, on page 11 the paper says "the domestic and international 'dual cycles' play an important role in promoting HQED." What does this mean? I interpret it to mean that HQED is generated by both domestic and foreign demand. Is this right?
Also, why are these called cycles? Is there some fluctuation like business cycles? I don't see anything about actual cycles in the paper--only development.
While the paper talks about dual cycle development, it does not directly discuss any development policies. Tables 1 and 2 list a number of economic variables, but they do not appear to be directly connected to any policies. The paper's conclusion that dual cycles promote high-quality economic development says nothing about how those dual cycles might be encouraged, or how they occur. The conclusion could be clearer in explaining the implications of the paper's empirical findings.
Table 1 has weights for each dimension. How are these weights calculated?
Author Response
1.Despite its frequent use of the term, the paper is not clear on what double-cycle development means. It appears to refer to domestic and international markets, but the paper should make this clear. The paper refers to industrial upgrading through the double cycle, but it is unclear what this means. Is it anything more than saying industry should be upgraded to meet demand? Some of that demand is domestic; some of it is foreign.
Response: According to the suggestions of experts, the paper introduces the double circulation in detail in the introduction part, and explains the promotion of industrial upgrading by double circulation. I am very grateful for the suggestions of the experts. The modified part is marked with red font.
2.As an example, on page 11 the paper says "the domestic and international 'dual cycles' play an important role in promoting HQED." What does this mean? I interpret it to mean that HQED is generated by both domestic and foreign demand. Is this right?
Response: Thanks to the reviewers for their comments. The explanation of the content is as follows: The domestic cycle as the main body aims to make full use of my country's complete industrial system, give full play to my country's huge market advantages and innovation potential, stabilize the industrial chain and economic operation, and effectively hedge against the growing international risks. The international cycle should focus on consolidating the production network with my country as the core platform, strengthen the integration with foreign industries, pay attention to the co-construction and mutual integration of the value chain of developed countries and the value chain of developing countries, and feed back the high-quality development of the domestic economy. The new development pattern of mutual promotion between domestic and international aims is to coordinate the overall situation of domestic and international. The synergy of two markets and two resources will provide more space for my country's economic development and industrial upgrading, and also create a relatively good environment for Chinese development.. The "dual circulation" strategy is the main measure to ensure the high-quality development of my country's economy. The modified part is marked with red font.
3.Also, why are these called cycles? Is there some fluctuation like business cycles? I don't see anything about actual cycles in the paper--only development.
Response: Thanks to the reviewers for their comments. The explanation of the content is as follows: "Dual circulation" refers to the two economic cycles of internal and external circulation, and building a new development pattern with the domestic cycle as the main body and the domestic and international dual cycles promoting each other.Emphasizing the main role of the domestic cycle means strengthening the effective operation of the domestic economic cycle, especially the interconnection of the five links of production, circulation, distribution, consumption and investment, and economic development mainly meets domestic demand. The mutual promotion of the two major cycles means that the domestic cycle is the foundation and foundation, and the international cycle is an extension and supplement. It's not a "closed country". It is necessary to strengthen the full utilization of domestic and international resources and jointly promote the high-quality development of China's economy. The modified part is marked with red font.
4.While the paper talks about dual cycle development, it does not directly discuss any development policies. Tables 1 and 2 list a number of economic variables, but they do not appear to be directly connected to any policies. The paper's conclusion that dual cycles promote high-quality economic development says nothing about how those dual cycles might be encouraged, or how they occur. The conclusion could be clearer in explaining the implications of the paper's empirical findings.
Response: Thanks to the reviewers for their comments. At present, the research is mainly at the exploratory stage. The economic variables selected in this paper are also condensed with reference to the strategies proposed by previous scholars and countries. For example, China proposed a dual-cycle strategy including five links: production, circulation, distribution, consumption, and investment. Variables are designed in these aspects. The high-quality development proposed by the state focuses on innovation, coordination, greenness and development. Therefore, this paper designs variables from this perspective. At the same time, a discussion of this aspect has been added in the empirical results section based on expert advice. Thanks for the expert opinion. The modified part is marked with red font.
5.Table 1 has weights for each dimension. How are these weights calculated?
Response: The paper explained in Section 3.3 that the weight calculation method is the entropy value method, which is not discussed due to space limitations. The modified part is marked with red font.
Reviewer 2 Report
The authors have done a great job in writing their article and build on an interesting and timely topic. The choice of research area and the methods used are appropriate and the paper will provide some very interesting theoretical and practical implications. There are however some points which you might want to improve in a minor revision.
First, I think it is important that you provide a clear and concise research question that drives your study in the introduction section. This will help a lot with the readability.
Second, it would be good to try to link your work with other studies that have focused on sustainable business strategies, particularly from the digital front. For example the paper of Kristoffersen et al., (2021) as well as that of Waqas et al., (2021) would be good to expand on and draw some links.
Third, it would be good to structure a bit the implications section and create sub-sections for the theoretical, practical implications, as well as for limitations and future work.
Kristoffersen, E., Mikalef, P., Blomsma, F., & Li, J. (2021). Towards a business analytics capability for the circular economy. Technological Forecasting and Social Change, 171, 120957.
Waqas, M., Honggang, X., Ahmad, N., Khan, S. A. R., & Iqbal, M. (2021). Big data analytics as a roadmap towards green innovation, competitive advantage and environmental performance. Journal of Cleaner Production, 323, 128998.
Author Response
The authors have done a great job in writing their article and build on an interesting and timely topic. The choice of research area and the methods used are appropriate and the paper will provide some very interesting theoretical and practical implications. There are however some points which you might want to improve in a minor revision.
First, I think it is important that you provide a clear and concise research question that drives your study in the introduction section. This will help a lot with the readability.
Response: Thanks to the reviewers for their suggestions.The introduction is reorganized according to the reviewer's suggestion, and the research of this paper is presented in a problem-oriented way. The modified part is marked with red font.
Second, it would be good to try to link your work with other studies that have focused on sustainable business strategies, particularly from the digital front. For example the paper of Kristoffersen et al., (2021) as well as that of Waqas et al., (2021) would be good to expand on and draw some links.
Response: I am very grateful for the high-quality reference materials provided by the experts. The author has read them carefully, and referred to some points of view, and cited them in the text. The modified part is marked with red font.
Third, it would be good to structure a bit the implications section and create sub-sections for the theoretical, practical implications, as well as for limitations and future work.
Response: According to the suggestions of the review experts, the meaning part, the suggestions, limitations and future work have been added. The modified part is marked with red font.
Kristoffersen, E., Mikalef, P., Blomsma, F., & Li, J. (2021). Towards a business analytics capability for the circular economy. Technological Forecasting and Social Change, 171, 120957.
Waqas, M., Honggang, X., Ahmad, N., Khan, S. A. R., & Iqbal, M. (2021). Big data analytics as a roadmap towards green innovation, competitive advantage and environmental performance. Journal of Cleaner Production, 323, 128998.
Reviewer 3 Report
Thank you the opportunity to revise the paper titled: “Exploring the Impact of “Double Cycle” and Industrial Upgrading on Sustainable High-Quality Economic Development: Application of Spatial and Mediation models”.
First of all I would like to congratulate the authors for a well written paper, dealing with an interesting topic. I have some comments that I hope the authors find useful:
Please check for some typos in the text, for instance in the abstract the words “seclects” or “regoins”. Similarly please check the spelling of capital letters, for instance “models; Next part” in line 79.
In the sentence of the intro: First, it analyzes the relationship between them in developing countries, what is “them”? It is not totally clear.
As the paper claims to contribute to the literature on developing countries, I would recommend adding a sub-section in the literature review devoted to developing/emergin countries to put this in context.
I would strongly recommend placing the sample description, and then the variables, before the estimation model description in the methodology section.
I would recommend naming the last section either Discussion, or Conclusion, but it is unclear why Discussion and Implication what is the difference between the two, and why implication in singular when there are several.
Good luck with your research!
Author Response
Thank you the opportunity to revise the paper titled: “Exploring the Impact of “Double Cycle” and Industrial Upgrading on Sustainable High-Quality Economic Development: Application of Spatial and Mediation models”.
1.First of all I would like to congratulate the authors for a well written paper, dealing with an interesting topic. I have some comments that I hope the authors find useful:
Please check for some typos in the text, for instance in the abstract the words “seclects” or “regoins”. Similarly please check the spelling of capital letters, for instance “models; Next part” in line 79.
Response: I am very grateful to the reviewers for their careful review. This revision carefully checked the language spelling and grammar problems of the paper, and made corrections to avoid language problems. The modified part is marked with red font.
2.In the sentence of the intro: First, it analyzes the relationship between them in developing countries, what is “them”? It is not totally clear.
Response: I am very grateful for the opinions of the experts. This revision has carefully checked the relevant issues. This article mainly studies the problems of the Yangtze River Economic Belt in China, and does not involve the problems of developing countries, and the direction of this part is not clear, so this part of the content is deleted. The modified part is marked with red font.
3.As the paper claims to contribute to the literature on developing countries, I would recommend adding a sub-section in the literature review devoted to developing/emergin countries to put this in context.
Response: I am very grateful for the opinions of the experts. Since the previous writing of the paper was a misunderstanding, this time it has been revised to remove the relevant words that affect developing countries, and strive to accurately reflect the research content of this paper. I would like to thank the experts for their careful review. The modified part is marked with red font.
4.I would strongly recommend placing the sample description, and then the variables, before the estimation model description in the methodology section.
Response: According to the suggestions of experts, this revision adjusts the description order of the method section, first the sample description, then the model construction, and finally the variable selection and description. The modified part is marked with red font.
5.I would recommend naming the last section either Discussion, or Conclusion, but it is unclear why Discussion and Implication what is the difference between the two, and why implication in singular when there are several.
Response: According to the suggestions of experts, this revision changed the title to Conclusion, and added suggestions, research shortcomings and outlook in the conclusion section. The modified part is marked with red font.
Reviewer 4 Report
Thank you for the opportunity of reading and reviewing your interesting manuscript. It addresses a highly relevant topic and uses spatial and mediation models for investigation. The paper is well written and well structured, although it has several shortcomings and it could be improved by considering the following suggestions:
1.please clarify the aims of the research, there is only indirect reference to the objectives;
2.the literature section / theoretical background is quite weak and it needs to be enhanced by referencing more relevant and novel titles;
3.the paper will increase its value if you formulate research hypotheses and then validate them;
4.the discussion of your findings could be performed more in-depth, and maybe a dedicated section can be introduced.
Good luck!
Author Response
Thank you for the opportunity of reading and reviewing your interesting manuscript. It addresses a highly relevant topic and uses spatial and mediation models for investigation. The paper is well written and well structured, although it has several shortcomings and it could be improved by considering the following suggestions:
1.please clarify the aims of the research, there is only indirect reference to the objectives;
Response: According to the suggestions of review experts, the introduction is reorganized, and the research of this paper is put forward in a problem-oriented manner, and the research purpose of this paper is clarified. The modified part is marked with red font.
2.the literature section / theoretical background is quite weak and it needs to be enhanced by referencing more relevant and novel titles;
Response:According to the suggestions of review experts, the literature section has been revised, and a large number of foreign literature and appropriate titles have been added. The modified part is marked with red font.
3.the paper will increase its value if you formulate research hypotheses and then validate them;
Response:Based on the recommendations of reviewers, the paper proposes hypotheses and validates them in the paper. The modified part is marked with red font.
4.the discussion of your findings could be performed more in-depth, and maybe a dedicated section can be introduced.
Response:According to the suggestions of the review experts, the paper discusses the research results, which improves the depth of the paper. I am very grateful for the suggestions of the experts. The modified part is marked with red font.
Round 2
Reviewer 1 Report
The revised manuscript clarifies the paper's ideas. The paper seems to come at the idea of a dual cycle from two directions. On the one hand, it provides empirical evidence on the effectiveness of dual cycle development. On the other hand, it seems to recommend dual cycle development. This is not necessarily inconsistent. The message seems to be "here is evidence that dual cycle development works, so the recommendation is to keep doing it." The paper could make this clearer.
Reviewer 3 Report
I have no further comments